# *Lactobacillus pentosus* MJM60383 Inhibits Lipid Accumulation in *Caenorhabditis elegans* Induced by *Enterobacter cloacae* and Glucose

**DOI:** 10.3390/ijms24010280

**Published:** 2022-12-23

**Authors:** Mingkun Gu, Pia Werlinger, Joo-Hyung Cho, Nari Jang, Shin Sik Choi, Joo-Won Suh, Jinhua Cheng

**Affiliations:** 1Interdisciplinary Program of Biomodulation, Myongji University, Yongin 17058, Republic of Korea; 2Myongji Bioefficacy Research Center, Myongji University, Yongin 17058, Republic of Korea; 3Department of Food and Nutrition, Myongji University, Yongin 17058, Republic of Korea

**Keywords:** lipid accumulation, antimicrobial activity, *Enterobacter cloacae*, *Lactobacillus pentosus*, probiotics

## Abstract

Gut microbiota are known to play an important role in obesity. *Enterobacter cloacae*, a Gram-negative bacterium, has been considered a pathogenic bacterium related to obesity in the gut. In this study, we established an obesity model of *C. elegans* by feeding *E. cloacae* combined with a high glucose diet (HGD), which significantly induced lipid accumulation. An anti-lipid mechanism study revealed that the fatty acid composition and the expression level of fat metabolism-related genes were altered by feeding *E. cloacae* to *C. elegans* under HGD conditions. Lactic acid bacteria that showed antagonistic activity against *E. cloacae* were used to screen anti-obesity candidates in this model. Among them, *L. pentosus* MJM60383 (MJM60383) showed good antagonistic activity. *C. eleans* fed with MJM60383 significantly reduced lipid accumulation and triglyceride content. The ratio of C18:1Δ9/C18:0 was also changed in *C. elegans* by feeding MJM60383. In addition, the expression level of genes related to fatty acid synthesis was significantly decreased and the genes related to fatty acid β-oxidation were up-regulated by feeding MJM60383. Moreover, MJM60383 also exhibited a high adhesive ability to Caco-2 cells and colonized the gut of *C. elegans*. Thus, *L. pentosus* MJM60383 can be a promising candidate for anti-obesity probiotics. To the best of our knowledge, this is the first report that uses *E. cloacae* combined with a high-glucose diet to study the interactions between individual pathogens and probiotics in *C. elegans*.

## 1. Introduction

Obesity can cause many diseases, such as type 2 diabetes, hypertension, cardiovascular disease, and arteriosclerosis due to a combination of genetic factors, fat metabolism disorders, dietary habits, and cultural factors. Consumption of a high-fat diet (HFD) is an important factor leading to obesity. Long-term intake of HFD can lead to the accumulation of fat in animals [1]. Various dietary plans, medications, and surgical procedures have been developed to control obesity. However, there is a need to develop safe and effective anti-obesity therapies to help weight management.

It was discovered that gut microbiota are associated with obesity [2]. Compositions of gut microbiota are different between obese and lean humans. Several studies have shown a decrease in Bacteroidetes but an increase in Firmicutes in obese people [3], suggesting that the HFD diet can influence gut microbial profile and diversity [4]. Interestingly, *Enterobacter cloacae* was recently discovered to be rich in obese humans [5]. It has been reported that *Enterobacter cloacae* B29 in combination with a high-fat diet, but not B29 alone, can cause obesity in germ-free mice, suggesting a pathogenic role of B29 in fat metabolism [5]. *Enterobacter cloacae* can produce lipopolysaccharide (LPS) that can stimulate inflammation in macrophages, thereby impairing colonic epithelial permeability and inducing obesity [6,7]. Most studies related to *E. cloacae* and obesity have either used germ-free mouse models or explored the distribution of *E. cloacae* in the gut of obese people. However, there are no studies on the use of *E. cloacae* to induce obesity in *C. elegans*.

*Caenorhabditis elegans* conserves 65% of the genes associated with human disease, and it has been widely used as an animal model for studies involving aging, obesity, and other physiological processes due to its relatively short lifespan, fast turnover, ease of maintenance, well-known genetic pathways, and greatly reduced experimental time and cost [8]. Studies have shown that the regulatory pathway for energy homeostasis is largely conserved from mammals to nematodes, including insulin signaling, fatty acid synthesis, β-oxidation, and serotonergic pathways [9]. Therefore, *C. elegans* offers great potential as an in vivo model for isolating and identifying compounds that modulate fat metabolism. More importantly, compared to rodent models, it is easier to study the function of individual gut bacteria due to the germ-free larvae.

In *C. elegans*, major fat is stored in intestinal and skin-like epidermal cells, which can be stained with lipid-affinity dyes such as Oil Red O (ORO), Sudan Black, and Nile Red. It was reported that over 300 genes are related to fat reduction, and 100 genes are associated with fat accumulation. For example, the gene *PEPT-1* (peptide transporter) affects fatty acid absorption by interacting with a sodium-proton exchanger NHX-2 to maintain the intracellular pH in the intestine of *C.elegans* [10]; the loss of *PEPT-1*promotes body fat accumulation. The genes *fat-6* and *fat-7* are involved in the fatty acid synthesis and can convert saturated fatty acids to monounsaturated fatty acids [11]. Gene *nhr-49* affects the expression of at least 13 genes involved in fatty acid desaturation, fatty acid β-oxidation, and fatty acid binding/transport. The deletion of *nhr-49* results in fat accumulation and the reduction expression of mitochondrial β-oxidation genes [8]. Gene *tub-1* is extensively expressed in the central nervous system. The loss of *tub-1* elevates fat accumulation [12]. Gene *acs-2* encodes a mitochondrial acyl-CoA synthetase, which facilitates the β-oxidation resulting in the degradation of stored fats [13].

Lactic acid bacteria (LAB) are important compositions of the gut microbiota. They showed many beneficial functions by inhibiting the growth of pathogenic bacteria, enhancing intestinal function, reducing metabolic disorders, and regulating the immune system [14,15,16,17,18]. Notably, lactic acid bacteria have shown anti-obesity effects, such as contributing to weight loss by modulating gut microbiota, regulating fat metabolism, lowering blood sugar levels, and inhibiting inflammation [19]. However, the studies on LAB with an anti-obesity activity using *C. elegans* are very limited. Therefore, this study aimed to establish a new obesity model by feeding *E. cloacae* to *C. elegans* and also screening LAB with anti-obesity activity by using this model. The anti-obesity activity of *Lactobacillus pentosus* MJM60383 and its mechanism of anti-lipid accumulation was evaluated and elucidated.

## 2. Results

### 2.1. E. cloacae Combined with a High Glucose Diet-Induced Lipid Accumulation in C. elegans

To establish an obesity model, fat accumulation was measured in the worms by feeding them with *E. cloacae* in the absence or presence of glucose. As shown in Figure 1A, the intensity of ORO dye was stronger in the HGD-OP50 group and HGD-*E. cloacae* group than in the ND-OP50 group. The lipid accumulation was increased by 2.1 fold in the HGD-OP50 group and 3.2 fold in HGD-*E. cloacae* group compared to the ND-OP50 group (Figure 2B). However, feeding *E. cloacae* without glucose supplementation didn’t increase lipid accumulation. The length and width of the worms in the HGD-*E. cloacae* group were also longer and wider than the other groups (Figure 2C,D).

### 2.2. Effects of E. cloacae Combined with High Glucose on Fatty Acids Composition and Expression of Fat Metabolism-Related Genes

To determine how *E. cloacae* combined a high glucose-induced fat accumulation in *C. elegans*, the total lipids and the fatty acids composition in each group were analyzed. In the HGD-*E. cloacae* group, the proportion of C18:0, C18:1 n9, C18:2 n6, C20:2, C20:3 n6, C20:4 n6, C22:0, C20:5 n3 were significantly increased compared to the HGD-OP50 group (Figure 2A). The desaturation index calculated by the ratio of C18:1Δ9/C18:0 was also increased by 20% compared to the HGD-OP50 group (Figure 2B).

We also determined the expression of genes related to fat metabolism in the HGD-OP50 and HGD-*E. cloacae* group. In the HGD-OP50 group, the gene expression of *asc-2* and *nhr-49* were significantly reduced compared to the ND-OP50 group, whereas *fat6* and *fat*7 were significantly upregulated in the HGD-OP50 group (Figure 2C). However, in the HGD-*E. cloacae* group, the gene expression of *fat6*, *pept-1*, *tub-1* and *fat7* were significantly increased, whereas the *acs-2* gene was significantly down-regulated compared to the HGD-OP50 group (Figure 2D).

### 2.3. Antagonistic Activity of LAB

A total of 70 strains isolated from fermented food were tested for their antagonistic activity against *E. cloacae*. Among them, 30 strains showed antagonistic activity (Table 1). Strains *L. pentosus* MJM60298, *L. plantarum* MJM60335, and *L. pentosus* MJM60383 showed stronger activity than LGG.

### 2.4. Screening of LAB with Anti-Lipid Accumulation Activity by Using HGD-E. cloacae Model

To test whether the anti-lipid accumulation activity of LAB is related to its antagonistic activity against *E. cloacae*, six strains (MJM60335, MJM60383, MJM60298, MJM60chosenMJM60454, MJM60556) were chosen to test their anti-obesity activity using the HGD-*E. cloacae* model. As shown in Figure 3A, the synchronized worms were grown on NGM plates seeded with OP50 or *E. cloacae* in the presence of glucose (100 mM) for 37 h at 20 °C, then they were transferred onto new NGM plates seeded only with LAB in the presence of glucose. After 18 h, the 55h-old adult worms were harvested to quantify their lipid accumulation by Oil Red O. In our study, the worms did not consume LAB from stage L1-L3 larvae worms.

Among the six strains, MJM60383, MJM60298, and MJM60335 reduced lipid accumulation by 42%, 16%, and 1%, respectively (Figure 3B). However, MJM60556, MJM6,0454, and MJM60494 did not show activity. Therefore, strain MJM60383 was selected for further study due to its high lipid-lowering activity.

### 2.5. MJM60383 Reduced Fat Deposition and TG Content Only in HGD-E. cloacae Model

To determine whether the lipo-reducing activity of MJM60383 is related to the *E. cloacae* diet, the fat accumulation in the HGD-OP50 model was also determined. Oil Red O staining and quantification showed that feeding of MJM60383 significantly decreased lipid accumulation in the HGD-*E. cloacae-induced* obesity model, but not in the HGD-OP50 model (Figure 4A,B). The feeding of MJM60383 decreased lipid accumulation by 58% in the HGD-*E. cloacae* model. However, fat deposition was not significantly changed by the feeding of MJM60383 in the HGD-OP50 model or the ND-OP50 model (Figure 4B).

In addition, the TG content of worms was reduced by 54% in the HGD-*E. cloacae*-MJM60383 group (Figure 4C). Moreover, the width of the worms (Figure 4D), but not the length, (Figure 4E) was significantly inhibited compared with HGD-*E. cloacae* group.

### 2.6. MJM60383 Altered Desaturation Index and Gene Expression

The fatty acid composition and the expression of the genes related to fat metabolism were altered in *C. elegans* fed with *E. cloacae* combined with a high glucose diet (Figure 2A,D). Then, the effect of diet change (feeding of MJM60383) on the ratio of C18:1Δ9/C18:0 and the expression of genes related to fat metabolism was investigated. We found that the feeding of MJM60383 can significantly decrease the ratio of C18:1Δ9/C18:0 by 30% compared to the HGD-*E. cloacae* group (LAB non-feeding control) (Figure 5A). In addition, in the MJM60383-feeding group, *fat6*, *pept-1*, *tub-1 and fat7* genes were significantly decreased, whereas *acs-2* and *nhr-49* genes were significantly increased compared to the HGD-*E. cloacae* group (Figure 5B).

### 2.7. Adhesion of LAB to the Caco-2 Cell and Bacterial Colonization in the Gut of C. elegans

To determine the adhesion ability of LAB to the gut cells, LAB was co-cultured with Caco-2 cells and the percent of attached cells was calculated, as shown in Figure 6. Strain MJM60335, MJM60298, ad MJM60383 showed an adherence of 5%, 20%, and 38% to the Caco-2 cell monolayer, respectively (Figure 6A).

To confirm whether MJM60383 can colonize in the gut of *C. elegans,* the CFU of bacteria (*E. cloacae* and MJM60383) was counted. We found that both *E. cloacae* and MJM60383 can colonize in the gut of nematodes. In addition, the CFU of *E. cloacae* in the gut of worms was significantly decreased by feeding MJM60383 compared with the control group (Figure 6B).

## 3. Discussion

Gut microbiota have been thought to be a very important factor in obesity development [2]. Recently, the administration of probiotics has been reported to reduce obesity in mice and humans. However, the mechanism has not been elucidated [20]. A recent study has shown that large amounts of Gram-negative bacteria are present in obese people [21]. These large groups of Gram-negative bacteria can produce high amounts of lipopolysaccharides (LPS) and trigger inflammation in humans and mice [22,23,24,25]. Notably, *E. cloacae*, an LPS-producing Gram-negative bacteria, was isolated from the gut of an obese human and caused obesity in germ-free mice [5]. *E. cloacae* can weaken the colonic epithelial barrier and cause obesity through an inflammatory response induced in LPS-stimulated macrophages [6,7]. Therefore, *E. cloacae* is considered to be an obesity-related pathogenic bacteria in the gut. To study the mechanism of the anti-obesity of probiotics, we established an obesity model by feeding *C. elegans* with *E. cloacae* and tested the anti-obesity effect of lactobacilli by establishing an induced obesity model.

Currently, an obesity model in *C. elegans* was established by adding 100 mM of glucose to the NGM medium. Peng H, et al. reported that lipid accumulation was increased by 30% in the HGD group compared to the normal-diet group [24]. In addition, in their study, *gene fat6* and *fat7* were significantly increased the in HGD model. However, genes *nhr-49* (Nuclear Hormone Receptor *NHR-49*) and *acs-2* were significantly downregulated [25]. Genes *fat6* and *fat7* can use stearic acid(C18:0) as a substrate to convert to oleic acid (C18:1Δ9). The gene *nhr-49* encodes a transcription factor involved in the control of pathways to regulate fat consumption and maintain the normal balance of fatty acid saturation by modulating the expression of genes involved in fatty acid beta-oxidation. The deletion of gene *nhr-49* increased fat accumulation in *C. elegans* [13]. *Acs-2*, one of the downstream genes of *nhr-49*, is an acyl-CoA synthetase catalyzing the conversion of fatty acid to acyl-CoA which is involved in fatty acid β-oxidation, resulting in a reduction of fatty acids [26]. The gene expression in the HGD model is consistent with our results (Figure 2C).

To understand the effect of LAB on *E. cloacae*-related obesity, an obesity model was established by using *E. cloacae* combined with a high-glucose diet (HGD-*E. cloacae*). In this model, worms were fed with *E. cloacae* instead of *E. coli* OP50, the standard diet for *C. elegans*. As shown in Figure 1, fat accumulation in the HGD-*E. cloacae* model was significantly increased compared to the HGD-OP50 group. Moreover, the length and width of worms were significantly increased in the HGD-*E. cloacae* model. However, worms fed with *E. cloacae* without glucose supplementation did not show an increase in fat accumulation. The results of *C. elegans* are consistent with that in mice reported by Fei, N., Zhao et. al [5]. They reported that *E. cloacae* can produce endotoxin under high-fat diet (HFD) conditions to induce inflammation and contribute to obesity in germ-free mice. However, *E. cloacae* can’t produce endotoxin under normal conditions, so it did not induce obesity. The high-fat diet was also reported to impact bacterial metabolism. Yoshimoto Et Al. reported that HFD altered gut microbiota, leading to an increase in deoxycholic acid, a gut bacterial metabolite known to cause DNA damage [27]. Similarly, Schwiertz et al. found significant differences in SCFA concentrations between lean and obese individuals [28]. Thus, we hypothesized that endotoxin producer *E. cloacae* could induce obesity in *C. elegans* under HGD conditions. In addition, we observed that some milky viscous substances were secreted by *E. cloacae* on the NGM plate (supplied with glucose). We suspected that these substances (water-soluble) might contribute to obesity induction. However, there are no reports indicating that the secretion of *E. cloacae* that can induce obesity, and whether these substances induce obesity needs further study.

Gut microbiota play a role in the modulation of genes associated with fat storage; specific diets have different capacities to regulate fat metabolism. This includes their micronutrient content, which can impact an organism. Micronutrients are vitamins and trace elements used as metabolites or cofactors in metabolic pathways [29]. In the previous study, genes *nhr-49* and *acs-2* were significantly downregulated in HGD-induced worms, which is consistent with our results. After feeding *E. cloacae* to *C. elegans*, the gene expression of *acs-2* was significantly inhibited compared to the HGD-OP50 group. Moreover, the genes *fat 6*, *fat 7*, *pept-1*, *and tub-1* were significantly upregulated compared to the HGD-OP50 group. Genes *fat 6* and *fat7* encode stearoyl-CoA desaturase (SCD), which is known to convert stearic acid (C18:0) to oleic acid and control the relative abundance of saturated and monounsaturated fatty acids in *C. elegans* [11]. These results suggested that the expression of fat metabolism-related genes is different between the HGD-OP50 and HGD-*E. cloacae* group. This result indicates that *E. cloacae* has different capacities to regulate fat metabolism from the *E. coli* OP50. In the HGD-*E. cloacae* group, the proportion of fatty acids (from C18 to C20) and the ratio of C18:1Δ9/C18:0 were significantly increased compared with the HGD-OP50 group. These results suggested that feeding *E. cloacae* to *C. elegans* under HGD conditions might increase fatty acid synthesis in *C. elegans* by regulating the expression level of the *fat6* and *fat7 genes*.

*Tub-1*, the TUBBY homolog, functions with *kat-1*, a 3-ketoacyl-CoA thiolase, to modulate fat metabolism, which is linked to fatty acid β-oxidation [30,31,32]. In addition, RAB-7 (Ras-related protein Rab-7a homolog), a mediated endocytic pathway, might be a target for tubby to regulate fat storage [30,31,32]. In the present study, the expression of the *tub-1* gene was increased in the HGD-*E. cloacae* model, suggesting a possible role of *E. cloacae* in fatty acid β-oxidation and the endocytic pathway. Along with *tub-1*, *nhr-49*, a functional homolog of PPARα, can also regulate β-oxidation [13]. However, the current results showed that *E. cloacae* did not influence *nhr-49*, whereas a downstream gene *acs-2* was significantly down-regulated by *E. cloacae*. Although both *nhr-49* and *tub-1* can modulate fatty acid β-oxidation, *E. cloacae* can modulate tub-1 but not *nhr-49.* Further studies are needed to determine the role of *E. cloacae* in tub-1-mediated fatty acid β-oxidation.

*PEPT-1*, an intestinal peptide transporter, can regulate fat deposition in the intestine of worms [33]. Previous studies have elucidated that *PEPT-1* affected fatty acid absorption by interacting with a sodium-proton exchanger NHX-2 to sustain the intracellular pH of intestinal epithelial cells [34]. In our study, the gene expression of *PEPT-1* was significantly increased in the HGD-*E. cloacae* group. An *E. cloacae* diet might modify the intracellular microenvironment. Taken together, these results suggest that *E. cloacae* combined with a high glucose diet might have induced fat accumulation in mainly upregulated fat metabolism-related genes (*fat6*, *fat7*, *acs-2*). Based on these results, an obesity model feeding *E. cloacae* to *C. elegans* under HGD conditions was established. To the best of our knowledge, this is the first report that *E. cloacae* combined with high glucose can induce fat accumulation in *C. elegans*.

To screen the *lactobacilli* with anti-lipid accumulation activity by using HGD-*E. cloacaea* model, we first tested the antimicrobial activity of LAB against *E. cloacae* (Table 2). The strains *L. pentosus* MJM60383, *L. pentosus* MJM60298, and *L. plantarum* MJM60335 showed strong antagonistic activity, which is higher than LGG. Other strains showed moderate activity against *E. cloacae*. Therefore, strains that showed strong or moderate activity were selected to test their anti-lipid accumulation activity in an HGD-*E. cloacae* induced obesity model of *C. elegans* (Figure 3). Among these tested strains, MJM60383 and MJM60298 significantly reduced lipid accumulation in the HGD-*E. cloacae* induced-obesity model. Although MJM60335 also showed strong antagonistic activity against *E. cloacae*, MJM60335 did not show inhibitory activity in HGD-*E. cloacae* induced obese *C. elegans*. Both MJM60383 and MJM60298 are identified as *L. pentosus* by the 16S rRNA sequence, and MJM60383 showed stronger activity than MJM60298, which indicated that this activity was strain-specific and not species-specific.

Strain MJM60383 only can reduce lipid accumulation in an HGD-*E. cloacae* induced-obesity model instead of an ND-OP50 and HGD-OP50 model (Figure 4B). However, the lipid-reducing mechanism of *L. pentosus* MJM60383 under the HGD-*E. cloacae* model should be studied further. To explore the difference in inhibitory activity between MJM60383 and other LAB with high antagonistic activity, the adhesion ability of LAB to Caco-2 cells was investigated due to major fat stored in intestinal and skin-like epidermal cells in *C. elegans*. Caco-2 cell was derived from human colon tissue and has been extensively used as a model for the study of probiotics to colonize the intestinal tract and compete with intestinal pathogens for attachment sites in the gut [35]. MJM60383 showed superior adherent activity to Caco-2 cells compared with MJM60298 and MJM60335. This result demonstrated that MJM60383 might have a high possibility of colonization in the gut of worms. It is necessary to consider comprehensive factors when selecting a probiotic candidate sin. We found that MJM60383 can colonize the gut of worms and reduce the CFU of *E. cloacae* in the gut of *C. elegans.* This result demonstrated that MJM60383 can compete with *E. cloacae* for the binding sites, leading to the elimination of the f *E. cloacae* from the gut or inhibition of the growth of *E. cloacae*, which may contribute to its anti-lipid accumulation activity. However, both Caco-2 cells and the gut of *C. elegans* are different from the gut of humans; it is necessary to verify the colonization of MJM60383 in the human gut through a clinical study in the future.

The role of MJM60383 on fat accumulation was investigated at the gene level (Figure 5B). Feeding *L. pentosus* MJM60383 to worms can significantly upregulate the expression of the *acs*-2 and *nhr*-49 genes, enhancing fatty acid β-oxidation and reducing lipids. Moreover, feeding MJM60383 to *C. elegans* can significantly down-regulate the expression of *fat6* and *fat7* genes, thereby affecting fatty acid synthesis and reducing the synthesis of triglycerides, the main component of lipids. Taken together, the gene expression was downregulated by feeding MJM60383 in the HGD-*E. cloacae* model, which might be attributed to its colonization of the intestine. MJM60383 colonized in the gut to possibly exclude *E. cloacae* or inhibit the growth of *E. cloacae*. The ratio of C18:1∆9/C18:0 was also reduced by feeding MJM60383, which confirmed the inhibition of *fat6* and *fat7* genes (Figure 5A).

## 4. Materials and Methods

### 4.1. C. elegans Culture and Bacterial Growth Conditions

Wild-type N2 Bristol *Caenorhabditis elegans* was obtained from the Caenorhabditis Genetics Center. Worms were fed with *Escherichia coli* OP50 and raised at 20 °C on nematode growth medium (NGM) agar plates. LAB was cultured in deMan-Rogosa-Sharpe (MRS) media at 37 °C for 24 h under anaerobic conditions. *Enterobacter cloacae* KACC 11403 was obtained from the Korean Agricultural Culture Collection and grown on nutrient media for 24 h at 37 °C under anaerobic conditions.

### 4.2. Establishment of the Induced-Obesity Model by Feeding E. cloacae Combined with a High-Glucose Diet (HGD)

Gravid nematodes were treated with a bleach solution (1 N NaOH: bleach: H_2_O; 2.5:1:0.5) to break down the worm’s bodies and obtain the eggs. Harvested eggs were hatched in sterilized M9 buffer (1M KH_2_PO_4_, 0.5M Na_2_HPO_4_, 4M NaCl, MgSO_4_·7H_2_O, H_2_O/ L) at 20 °C for 16 h to obtain age-synchronized germ-free larvae worms (L1 larvae) [36]. L1 larvae worms were fed with *E. cloacae* in the absence or presence of glucose to establish an induced-obesity model. In the ND-OP50 group, synchronized worms were fed with OP50 on an NGM plate without glucose supplementation and incubated at 20 °C for 55 h. In the HGD-OP50 group, synchronized L1 worms were fed with OP50 on an NGM plate containing 100 mM glucose and incubated at 20 °C for 55 h. In the ND-*E. cloacae* group, synchronized L1 worms were fed with *E. cloacae* on an NGM plate without glucose supplementation and incubated at 20 °C for 55 h. In the HGD-*E. cloacae* group, synchronized L1 worms were fed with *E. cloacae* on an NGM plate supplemented with 100 mM glucose and incubated at 20 °C for 55 h.

### 4.3. Oil Red O Staining and Quantification

Oil Red O staining of *C. elegans* was performed according to the method described before [37]. Briefly, worms were washed from the plates with phosphate-buffered saline supplied with 0.01% Triton X-100 (PBST). After washing, the pellet of worms was treated with 40% isopropanol and incubated for 3 min at room temperature (RT). These worms were then resuspended in 60% Oil Red O solution (5 mg/mL) and rotated at 30 rpm for 2 h at RT. Worms were imaged with a Ci-L microscope and a color camera (Nikon Co., Japan).

To quantify lipid accumulation in the worms, stained worms were resuspended in M9 buffer containing 0.1% Tween-20, and the worm suspension (10 μL) was pipetted onto slides to count the number of worms. To obtain accurate worm numbers in each 10 μL suspension, we repeated the counting independently three times and took the mean. A total of 2000 stained worms were added to 200 μL isopropanol and incubated at 37 °C for 30 min. The lipid content was determined by measuring the dye extracted from the stained worms at an absorbance of 510 nm by a microtiter plate reader (Tecan, Austria).

### 4.4. Fatty Acids Composition

The fatty acid composition of worms in each group was determined according to the method of [25]. Briefly, *C. elegans* was washed with M9 buffer to remove diet bacteria. Worm lipids were extracted with 1 mL solvent (2% H_2_SO_4_ mixed with 98% methanol) at 80 °C for 2 h. After incubation, 1 mL of n-hexane was added and mixed. The solvent was then separated into two layers (methanol and hexane phases). The upper layer (hexane phase) was evaporated by nitrogen gas and then weighed. Each sample contained at least 5 mg of lipid.

The lipids of worms were analyzed by gas chromatography−mass spectrometry (GC-MS, Agilent, USA) using a DB-WAX column (120 mm × 0.25 mm × 0.25 μm). The oven temperature was programmed as follows: the initial temperature was 80 °C, held for 3 min, programmed to 200 °C at 15 °C/min, and finally maintained at 80 °C for 3 min. The fatty acids composition of *C. elegans* was identified by comparison of the retention time and mass spectra with the commercial standard of fatty acid. The result of fatty acids is expressed as the weight. The desaturation index was then determined by measuring the ratio of C18:1 Δ9/18:0 [38].

### 4.5. Antagonistic Activity of LAB

An antagonistic assay was performed to determine if lactobacilli isolated from fermented food have antimicrobial activity against *E. cloacae* KACC 11403 [39]. Briefly, a nutrient soft agar (0.7% agar) was seeded with 0.1% (*v/v*) *E. cloacae* (1 × 10^7^ CFU/mL) and overlaid on MRS agar. Holes were made on MRS plates using a borer. The overnight culture (10 μL) of lactic acid bacteria was poured into the hole on the MRS plate and incubated at 37 °C for 24 h. *Lactobacillus rhamnosus* GG (LGG) was used as a control. After incubation, the inhibition zone was measured.

### 4.6. Screening of LAB with Inhibitory Activity for Lipid Accumulation

The egg was synchronized as described above. After hatching, L1 worms were conditioned with *E. cloacae* for 37 h at 20 °C. The worms were then washed with M9 buffer to completely remove *E. cloacae* and transferred to a new NGM plate (supplemented with 100 mM glucose. A total of 2000 worms from each group were collected and quantified by lipid quantification as described above. To confirm that *C. elegans* consume LAB, worms were conditioned with *E. cloacae* for 37 h and transferred to an NGM blank without any bacterial lawn used as a blank group.

### 4.7. Triglyceride (TG) 

55-h-old adult worms fed with *E. cloacae* or LAB were harvested. The TG content in each group (containing 2000 worms) was determined by using a PicoSens™ Triglyceride Assay Kit (Biomix, Guri, Republic of Korea) according to the manufacturer’s instructions. Protein content in the supernatant was determined using a Pierce^TM^ BCA protein Assay kit following the manufacturer’s instructions. The triglyceride content was normalized by protein [40]. Relative TG content was calculated with the following formula:Relative TG content (100%) = TG_sample_/TG_model_ × 100(1)

### 4.8. RNA Extraction and Quantitative Real-Time PCR

To determine the influence of the bacterial diet on the expression of fat metabolism-related genes, 7000 adult worms from each group were washed with M9 buffer and centrifuged at 1200 rpm for 2 min. The worm pellet was frozen in liquid nitrogen and then thawed completely in a heating block at 40 °C. for five times to destroy the worm’s cuticle. For RNA isolation, the worm pellet was treated with 1 mL of Trizol reagent and rotated for 20 min at RT, then 200 µL of chloroform was added to the worm pellet and kept on ice for 15 min. Samples were centrifuged at 14,000 rpm for 15 min at 4 °C. The aqueous phase was transferred to a new microcentrifuge tube containing 500 µL isopropanol. After incubating on ice for 10 min, the mixture was centrifuged at 14,000 rpm for 15 min at 4 °C. After centrifugation, the supernatant was discarded and 1 mL of 75% cold ethanol was added to the pellet. After centrifuging at 12,000 rpm for 5 min at 4 °C, the supernatant was discarded and the RNA pellet was air-dried and dissolved in 30 µL RNase-free water.

For qRT-PCR quantification, RNA was reverse transcribed into cDNA with a PrimeScript^TM^ RT Reagent kit and gDNA Eraser (TAKARA Bio INC, Shiga, Japan) reagent according to the manufacturer’s protocol. A real-time quantitative RT-PCR was performed on a Roche Light Cycler^®^ 96 System (Roche, Basle, Switzerland) using a Maxime^TM^ PCR PreMix kit (LIliF, Seongnam, Republic of Korea). Primers used for each gene are listed in Table 2. *Act-1* was used as an internal control. Quantitative PCR conditions were as follows: one cycle at 95 °C for 2 min, followed by 40 cycles at 95 °C for 10 s, 53 °C for 10 s, and 72 °C for 20 s. The fold change was then calculated using the 2^-∆∆Ct^ method [41]. All experiments were performed in triplicate.

### 4.9. Adherence of LAB to Caco-2 Cell

The Caco-2 cell line was purchased from the American Type Culture Collection (ATCC). The cells were cultured in Dulbecco’s Modified Eagle’s minimal essential medium (DMEM; Gibco) supplemented with 10% (*v/v*) heat-inactivated fetal calf serum (Gibco), 2 mM l-glutamine (Sigma),100Uml^−1^ penicillin and 100 mg ml^−1^ streptomycin (Sigma) at 37 °C in 5% CO_2_ condition. For the adhesion assay, Caco-2 monolayers were seeded at a concentration of 5 × 10^5^ cells per well in 12-well plates. Cells were treated with LAB suspension (1 × 10 ^7^CFU/mL) for 2 h when the cells reached 80% confluence. After incubation, the adherent LAB bacteria and Caco-2 cells were detached by adding 1ml of 0.05% (*v/v*) Triton-X 100, and the CFU of adherent bacteria was calculated by serial dilution and spreading on MRS agar plates. The percentage of LAB adherence was calculated as follows:Adhesion (%) = (CFU of adherent bacteria/CFU of initial inoculated bacteria) × 100(2)

### 4.10. Counting of Bacteria in the Gut of C. elegans

The bacterial colonization in C. elegans was determined and described by Daniela Uccelletti et al. [42], with minor modifications. Worms (2000 worms in each group) either fed with E. cloacae under HGD conditions or fed with MJM60383 in the HGD-E. cloacae model were harvested at 56 h at 20 ℃ and then washed three times with M9 buffer containing 20 μg/mL gentamycin to remove surface MJM60383 and E. cloacae. Worm pellets were washed with PBS and cracked with 1 mL of M9 buffer–1% Triton X-100. Worm lysates were then diluted with a PBS buffer and plated on Nutrient agar (for E. cloacae) or MRS agar (for MJM60383) at 37 °C for 48 h, respectively. E. cloacae and L. pentosus MJM60383 colonies were quantified and used to calculate the number of bacteria. This experiment was performed in triplicate.

### 4.11. Statistical Analysis

All measurements were repeated independently in triplicate. Results are expressed as mean ± standard deviation (SD). The data obtained were statistically analyzed using GraphPad Prism version 8. A one-way analysis of variance was used to study the significant difference between means, with a significance level of * *p*  <  0.05, ** *p* < 0.01, *** *p* < 0.001, **** *p* < 0.0001.

## 5. Conclusions

To the best of our knowledge, this is the first report that *E. cloacae* combined with a high-fat diet can induce lipid accumulation in *C. elegans*. It provides a strategy to study the interaction between individual pathogens and probiotics, which is more effective and less time-consuming than rodent models. *L. pentosus* MJM60383 reduced lipid accumulation and triglyceride content and regulated the expression of lipid-related genes in *C. elegans* fed with *E. cloacae* combined with a high glucose diet, thus it can be a promising candidate for anti-obesity probiotics.

## Figures and Tables

**Figure 1 ijms-24-00280-f001:**
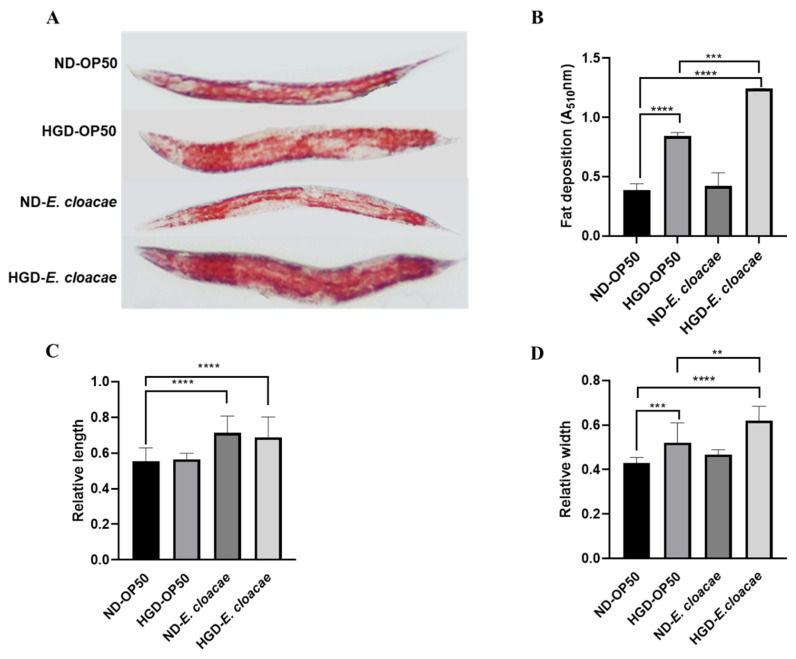
Establishment of the obesity model induced by feeding *E. cloacae* under HGD conditions. *C. elegans* (L1) was fed with *E. coli* OP50 or *E. cloacae* in the absence or presence of 100 mM glucose on the NGM plate and incubated at 20 °C for 55 h. (**A**) Oil Red O staining of *C. elegans*. (**B**) Quantification of Oil Red O. Data are expressed as means ± SD (n = 3, 2000 worms in each independent experiment). (**C**) The relative length of worms (n = 25). (**D**) The relative width of worms (n = 25). Statistically significant differences between the treated group and control group as analyzed using the one-way ANOVA with Dunnett’s multiple comparison test (** *p* < 0.01, *** *p* < 0.001, **** *p* < 0.0001).

**Figure 2 ijms-24-00280-f002:**
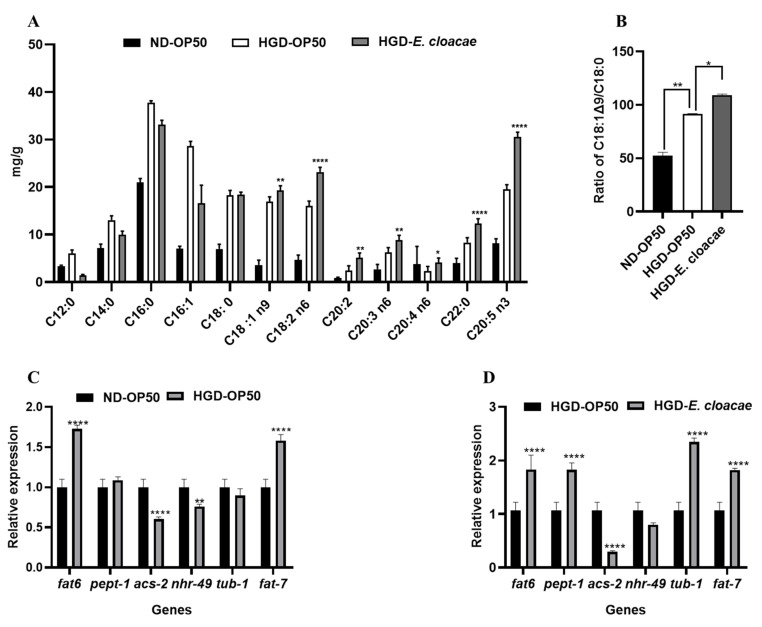
*E. cloacae* combined with a high glucose diet changed the fatty acids profile, the desaturation index, and the expression level of fat metabolism-related genes in *C. elegans.* (**A**) Fatty acids profiles of worms. (**B**) The desaturation index was measured by the ratio of C18:1 Δ9/C18:0 (n = 3). (**C**) Relative expression level of fat metabolism-related genes in HGD-induced model. (**D**) Relative expression level of fat metabolism-related genes in HGD-*E. cloacae* model. Statistically significant differences between the treated group and control group were analyzed using the one-way ANOVA with Dunnett’s multiple comparison test (* *p* < 0.05, ** *p* < 0.01, **** *p* < 0.0001).

**Figure 3 ijms-24-00280-f003:**
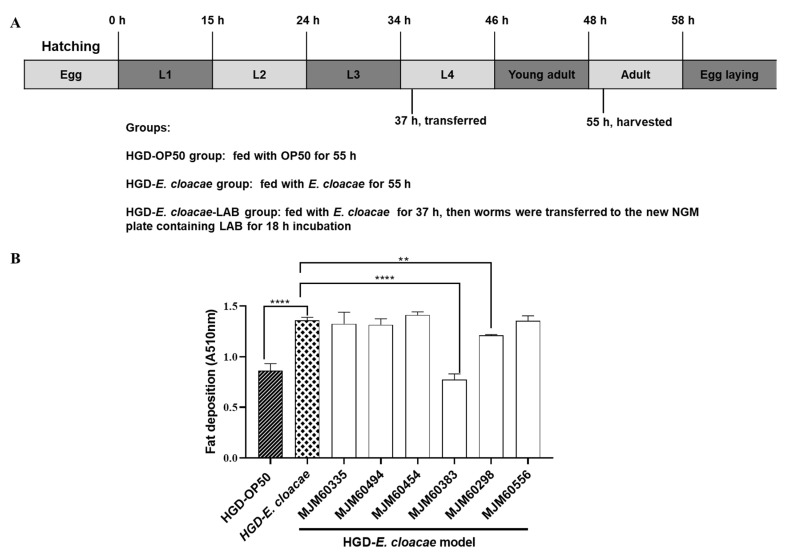
Screening of LAB with anti-lipid accumulation activity by using the HGD-*E. cloacae* model. (**A**) Experimental scheme of LAB feeding. Synchronized L1 worms were fed with OP50, or *E. cloacae* (NGM plate supplemented with 100 mM glucose) for 37 h, then worms were transferred onto new NGM plates (100 mM glucose) plated with LAB lawn and raised for 18 h at 20 °C. (**B**) Quantification of Oil Red O. Data are expressed as means ± SD (n = 3, 2000 worms in each independent experiment). Statistically significant differences between the treated group and control group were analyzed using the one-way ANOVA with Dunnett’s multiple comparison test (** *p* < 0.01, **** *p* < 0.0001 compared with the HGD-*E. cloacae* group).

**Figure 4 ijms-24-00280-f004:**
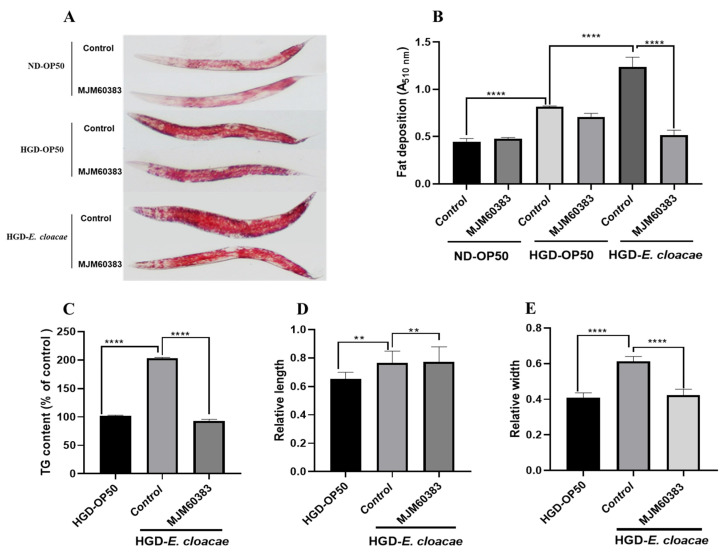
MJM60383 reduced the fat deposition and the TG content in the HGD-*E. cloacae* model. Synchronized L1 worms were fed with OP50, or *E. cloacae* (NGM plate supplemented with or without 100 mM glucose) for 37 h at 20 °C, then worms were transferred into new NGM plates supplemented with MJM60383 lawn and raised for 18 h at 20 °C. (**A**) Representative images of Oil Red O staining. (**B**) Quantification of ORO in each group. For Oil Red O quantification, the data were obtained from three independent experiments and 2000 stained worms were used in each experiment. (**C**) TG content in worms (n = 3, 2000 worms in each independent experiment). Triglyceride levels were normalized by protein content. (**D**) The relative length of worms (n = 25). (**E**) The relative width of worms (n = 25). Statistically significant differences between the treated group and control group were analyzed using the one-way ANOVA with Dunnett’s multiple comparison test (** *p* < 0.01, **** *p* < 0.0001).

**Figure 5 ijms-24-00280-f005:**
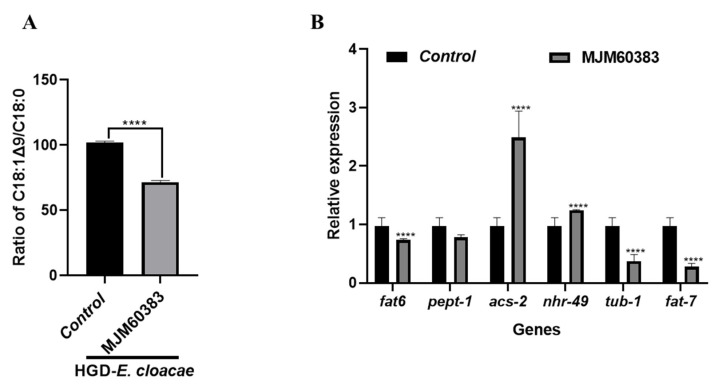
MJM60383 altered fatty acids composition and fat metabolism-related gene expression. (**A**) The desaturation index was measured by the ratio of C18:1Δ9/C18:0 (n = 3 samples, each sample containing 5 mg of lipid). (**B**) The relative expression level of fat metabolism-related genes. Data are expressed as the mean ± SD. Statistically significant differences between the treated group and control group were analyzed using the one-way ANOVA with Dunnett’s multiple comparison tests (**** *p* < 0.0001, compared with the HGD-*E. cloacae* group).

**Figure 6 ijms-24-00280-f006:**
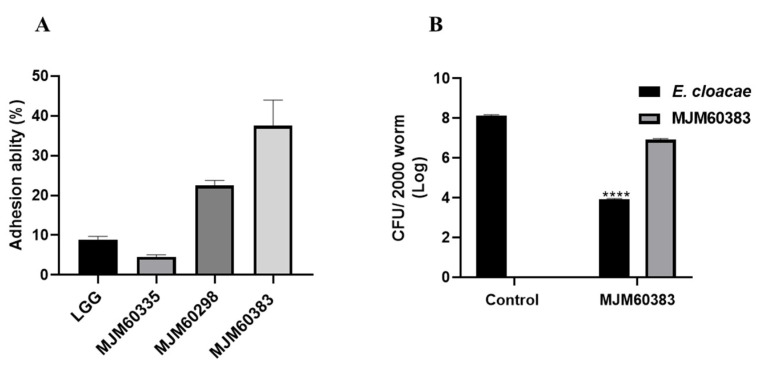
Cell adhesion activity of LAB and bacterial colonization in the gut of *C. elegans*. (**A**) Adherence ability of isolates to Caco-2 cell monolayer. Error bars represent the standard deviation of three independent experiments. (**B**) Bacterial colonization in the gut of *C. elegans.* In the control group, worms were fed with *Ecloacaee* only for 55 h. In the HGD-*E. cloacae* model, worms were conditioned with *E. cloacae* for 37 h, then transferred to an NGM plate seeded with MJM60383 and incubated for 18 h. Bacteria were calculated as Log CFU in each group (containing 2000 worms). **** *p* < 0.0001, Statistical significance was determined by a two-tailed unpaired Student’s t-test between two groups.

**Table 1 ijms-24-00280-t001:** Antagonistic activity of LAB against *E. cloacae*.

Strain	Species *	Inhibition Zone (mm)
LGG	*L. rhamnosus GG*	26
MJM60298	*L. pentosus*	28
MJM60335	*L. plantarum*	28
MJM60371	*Leuconostoc mesenteriodes*	2
MJM60376	*L. mesenteriodes*	21
MJM60381	*L. plantarum subsp*	19
MJM60383	*L. pentosus*	30
MJM60385	*L. rhamnosus*	18
MJM60390	*L. brevis*	18
MJM60395	*L. helveticus*	20
MJM60396	*L. paracasei*	18
MJM60397	*L. fermentum*	26
MJM60405	*Pediococcus pentasaceous*	17
MJM60419	*L. helveticus*	17
MJM60427	*L. acidophilus*	16
MJM60429	*L. plantarum*	25
MJM60432	*L. casei*	16
MJM60433	*L. casei*	23
MJM60434	*L. paracasei*	22
MJM60436	*L. zeae*	22
MJM60437	*L. helveticus*	22
MJM60439	*L. fermentum*	25
MJM60448	*L. rhamnosus*	25
MJM40494	*L. plantarum*	0
MJM60556	*L. sakei*	15
MJM60454	*L. rhamnosus*	16
MJM60456	*L. salivarius*	2
MJM60458	*L. reuteri*	18
MJM60559	*L. brevis*	2
MJM60460	*L. paracasei subsp.Paracasei*	17
MJM60461	*L. plantarum*	17
KACC12311	*Pediococcus pentosaceus*	15

* Identified by 16S rRNA sequence.

**Table 2 ijms-24-00280-t002:** Primers used for PCR and RT-PCR.

Gene	Primer Sequence (5’→3’)	Amplicon (bp)	Tm (°C)
*fat-6*	F: TCA ACA GCG CTG CTC ACT AT	170	54
	R: TTC GAC TGG GGT AAT TGA GG		
*fat-7*	F: CAA CAG CGC TGC TCA CTA	362	57
	R: CAC CAA CGG CTA CAA CTG		
*acs-2*	F: GCC TTG GAT GGG ATA GAG	122	52
	R: TGA TGG GAA GAC CAC AGT		
*tub-1*	F: CCA CAG CAA GTT CAA GAG TC	301	55
	R: AGC CAC TAC ATC AGT GTT CC		
*pept-1*	F: GTG TTC GGA GAA GTA TCT CG	176	56
	R: CAA GAG CAC AGT CGT GAG TA		
*nhr-49*	F: GCT CTC AAG GCT CTG ACT C	134	56
	R: GAG AGC AGA GAA TCC ACC T		
*act-1*	F: GAG CGT GGT TAC TCT TTC A	68	54
	R: CAG AGC TTC TCC TTG ATG TC		

F, forward; R, reverse.

## Data Availability

The raw data used to support the findings of this study will be made available by the authors, without undue reservation, to any qualified researcher.

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
