# Peer review of "Lactobacillus pentosus* MJM60383 Inhibits Lipid Accumulation in *Caenorhabditis elegans* Induced by *Enterobacter cloacae* and Glucose"

_ijms, 2022, doi:10.3390/ijms24010280_

Round 1

Reviewer 1 Report

The manuscript is nicely written to identify the promising candidate for the anti-obesity probiotics. I have several minor comments for authors.

First, the manuscript should follow the order: Introduction, Method, Results, Discussion and Conclusion. The current flow is not easy to read.

Second, the introduction is still too thin to justify the research question, especially for the last two paragraphs. What was the evidence to demonstrate the anti-obesity effects of LAB in different models, especially for C. elegans? In the last paragraph, there is a logic jump to talk about Lactobacillus pentosus MJM60383 all in a sudden. After the phrase “by using this model”, still a lot needs to be clarified.

Last but not least, authors will need to clear all typo throughout the manuscript, e.g. “inflammation” in line 83, as well as all inconsistencies in formatting, e.g. “(24-26)” in line 213 but “[5], [6]” in line 217.

Reviewer 2 Report

See attached PDF.

Round 2

Reviewer 1 Report

Authors have addressed my concerns adequately, I don't have further comments.

Reviewer 2 Report

I am satisfied with the revisions.